# Comparison of hematological parameters between type 2 diabetes mellitus patients and healthy controls at Dessie comprehensive specialized hospital, Northeast Ethiopia: Comparative cross-sectional study

**Hussen Ebrahim**[1]*, **Temesgen Fiseha**[1], **Yesuf Ebrahim**[2], **Habtye Bisetegn**[1]

**1** Department of Medical Laboratory Sciences, College of Medicine and Health Sciences, Wollo University, Dessie, Ethiopia, **2** Department of Medical Laboratory Sciences, Dessie Health Science College, Dessie, Ethiopia

* husshosam@gmail.com

## Abstract

### Background

Diabetes mellitus (DM) is a chronic condition associated with raised levels of blood glucose due to the body cannot produce any or enough insulin hormone or cannot be effectively utilized the produced insulin by the body. Patients with poorly controlled diabetes show a significant alteration in various parameters including metabolic, cellular, immunological, and hematological disturbances that leads to vascular complications. Thus, the main aim of this study is to compare hematological parameters between type 2 diabetes mellitus (T2DM) patients and healthy controls.

### Methods

A comparative cross-sectional study was conducted in Dessie comprehensive specialized hospital from January to June 2021. A total of 240 study participants consisting of 120 T2DM patients and 120 healthy controls were recruited using a systematic random sampling technique. Hematological parameters were determined using the DIRUI BF6500 automated hematology analyzer. Independent T-test was used to compare the mean of hematological parameters between T2DM patients and healthy controls. Pearson correlation test was used to determine the correlation between FBG, BMI, SBP, DBP, and hematological parameters in T2DM patients. Multivariate logistic regression was used to assess the association between socio-demographic and clinical variables with anemia. The result was expressed in mean and standard deviation and presented in texts and tables. P-value < 0.05 was considered to be statistically significant.

### Results

The mean and standard deviation of monocyte count, basophil count, monocyte %, basophil %, RBC count, hematocrit, MCV, MCH, RDW-SD, MPV, PDW, PLC-R, and plateletcrit

**Data Availability Statement:** All relevant data are within the manuscript and its Supporting Information files.

**Funding:** The author(s) received no specific funding for this work.

**Competing interests:** The authors have declared that no competing interests exist.

showed a significant difference between T2DM patients and healthy control group. Pearson correlation coefficient showed that the total WBC count, neutrophil count, monocyte count, basophil count, RDW-CV, PDW, MPV, PLC-R, and plateletcrit were statistically positively correlated with FBG whereas RBC count, Hgb, hematocrit, MCV, MCH, and RDW-SD were statistically negatively correlated with FBG in T2DM patients. Moreover, total WBC count, neutrophil count, monocyte count, basophil count, Hgb, and plateletcrit were statistically positively correlated with BMI while RBC count, Hgb, hematocrit, MCV, MCH, and RDW-SD were statistically negatively correlated with BMI in T2DM patients. On the other hand, DBP was significantly positively correlated with platelet count and RDW-CV whereas SBP also significantly positively correlated with total WBC count, neutrophil count, basophil count, and PDW. Besides, DBP and SBP showed statistically significant negative correlations with RBC count, Hgb level, and Hct value in T2DM patients. The overall prevalence of anemia was 25.8% in T2DM patients with a higher prevalence of anemia (16.7%) in female patients. Multivariate logistic regression revealed that being non-employee worker (AOR: 3.6, 95% CI, 1.4–46.0, P = 0.002), presence of neuropathy (AOR: 13.40, 95% CI, 6.83–26.28, P = 0.00), and duration of the disease $\geq$ 5 years (AOR = 3.2, 95% CI, 1.2–15.3, P = 0.03) have had statistically significant association with anemia inT2DM patients.

## Conclusions

Patients with T2DM may have significant alterations in various hematological parameters. Hematological parameters should be regularly tested for early diagnosis and proper management of diabetes-related complications.

## Introduction

Diabetes mellitus (DM) is a chronic condition associated with raised levels of blood glucose due to the body cannot produce any or enough of the hormone insulin or cannot effectively utilize the produced insulin by the body [1]. It is considered to be a heterogeneous group of metabolic disorders characterized by several defects in the regulation of carbohydrate, fat, or protein metabolism [2]. Diabetes mellitus is characterized by hyperglycemia, glycosuria, hyperlipidemia, and negative nitrogen balance in the body. Poorly or uncontrolled level of hyperglycemia could be associated with multiple organ damage leading to disabling and life-threatening health complications such as cardiovascular diseases (CVD), nerve damage (neuropathy), kidney damage (nephropathy), lower-limb amputation, and eye disease (mainly affecting the retina) resulting in vision loss and blindness [1, 2].

Diabetes can be classified into four general categories that include type 1 diabetes mellitus (T1DM) which is caused by autoimmune pancreatic beta-cell destruction usually leading to absolute insulin deficiency; type 2 diabetes mellitus (T2DM) which is caused by a progressive loss of adequate beta-cell insulin secretion frequently on the background of insulin resistance; other specific types of diabetes due to other causes (such as monogenic diabetes syndromes, diseases of the exocrine pancreas and drug or chemical-induced diabetes) and gestational diabetes mellitus is pregnancy-induced diabetes that could be diagnosed in the second or third trimester of pregnancy, not overt diabetes before pregnancy [1, 3]. Moreover, T1DM and T2DM could be related to multiple consequences because hyperglycemia could be associated with many physiological processes that may affect lipid metabolism, the regulation of inflammation,

vasodilatation, vascular, immunological, and hematological parameters, cell growth, and replication in poorly or uncontrolled diabetes [4].

Diabetes is considered to be the main risk factor for CVD. Independent risk factors that may include obesity, hypertension, toxins, and other genetic factors may be the possible risk factors for diabetes and CVD. Diabetes and CVD are significantly related to glucose intolerance, hyperinsulinemia, dyslipidemia, insulin resistance syndrome, inflammation, thrombophilia, dysglycemia, oxidative stress, inflammation, endothelial dysfunction, hemostatic and hematological abnormalities, and the generation of atherogenic lipoproteins [5–7]. Patients with poorly controlled diabetes may have a significant alteration in various parameters including metabolic, cellular, and hematological disturbances that leads to vascular complications [8–10]. Hematological alterations including the change in the function, structure, and metabolism of red blood cells (RBCs), white blood cells (WBCs), platelet count and platelet indices, and hemostatic parameters are the encountered abnormalities in T2DM patients. Anemia is one of the commonest hematological abnormalities that could be seen in patients with diabetes usually occurring earlier and to a greater degree in patients presenting with diabetic nephropathy [11].

According to the World Health Organization (WHO), anemia is defined as hemoglobin (Hgb) levels of less than 12.0 g/dl for women and less than 13.0 g/dl for men but, normal Hgb level varies not only with sex but also with ethnicity, physiological status and altitude [12]. Prolonged hyperglycemia may lead to increased production of reactive oxygen species (ROS) and the formation of advanced glycation end products (AGEs) which are directly associated with endothelial dysfunction and hematological changes. The increased production of oxidative stress caused by the excessive release of ROS may cause tissue damage and hematological alterations such as RBC dysfunction, platelet hyper activation, and endothelial dysfunction. Furthermore, insulin resistance is also associated with endothelial dysfunction, increased plasma level of inflammatory markers, increased WBC count, and platelet hyper activation that may trigger and accelerate vascular complications in T2DM patients [13, 14].

In diabetes, hematological changes are directly associated with endothelial dysfunction and inflammation. Hyperglycemia and its metabolic syndrome can be related to the alteration of the different hematological parameters such as the morphology, size, and physiological functions of RBCs, WBCs, and platelets [10]. Moreover, oxidative stress results in RBC dysfunction, platelet destruction, and tissue injury which may affect the function of blood cells and the hemostatic parameters that may lead to various complications [15]. There have been few studies conducted to investigate the alterations of hematological parameters among T2DM patients. Besides, hematological parameters may not be routinely determined as laboratory diagnostic biomarkers to monitor diabetes and diabetes-associated complications in developing countries specifically in Ethiopia. There was no similar study conducted in the study area to assess the alterations of hematological parameters among diabetes patients. Therefore, the main aim of this study was to compare the hematological parameters between T2DM patients and healthy controls.

## Methods and materials

### Study design, period, and area

A comparative cross-sectional study was conducted at Dessie comprehensive specialized hospital, Northeast Ethiopia from January to June 2021. The hospital is found in Dessie town which is about 400 Km far from the north of Addis Ababa, the capital of Ethiopia. The hospital gives service to more than four million people. The hospital provides emergency, antiretroviral therapy services, chronic care, surgical, dental, medical, pediatric, gynecologic, obstetric, and

other services for both inpatient and outpatient clients. Moreover, beyond clinical service, the hospital also provides teaching services for undergraduate, graduate, and specialty programs for different health professionals.

## Study population and participants

The source population for this study was all T2DM patients who came to the hospital for follow-up service and were included in the diabetic group and all healthy blood donors who came to Dessie blood bank for blood donation were included in the control group. The study was conducted on a total of 240 study participants consisting of 120 T2DM patients who attended the diabetes clinics and 120 healthy controls who attended to Dessie blood bank for blood donation that fulfilled the eligibility criteria were included in the study. For both diabetes and healthy controls, individuals who had a history of known inherited bleeding disorders, chronic renal disease, chronic liver disease, any history of malignancy, infectious diseases (human immunodeficiency virus (HIV), hepatitis B virus (HBV), and hepatitis C virus (HCV)), chronic smoker, regular chat chewer, chronic alcohol drinker, and pregnant women were excluded from the study.

## Sample size determination and sampling technique

The sample size was calculated by using the double population proportion or the double population means formula by considering a 95% confidence interval and 5% margin of error, and by taking the significant hematological parameters from previous studies. Unfortunately, using the above formula and taking the different hematological parameters as significant variables from different studies, the small sample size was too small to make inferences. So, the authors used a rule of thumb as an alternative means to determine the sample size. According to the rule of thumb that has been recommended by van Voorhis and Morgan has been applied, at least 30 study participants per group are required to detect real differences which could lead to about 80% power of the study if the study contains two or more groups and if we used to T-test, ANOVA, etc as statistical tools for analysis [16]. To maximize the reliability of the study, the investigator tried to increase the study participants proposed by the rule of thumb four-fold in each study group. Thus, a total of 240 study participants (120 T2DM and 120 age and gender-matched healthy controls) were enrolled in the study. A systematic random sampling technique was employed to recruit study participants attending the hospital during the study period.

## Data collection and laboratory analysis

Socio-demographic characteristics of study participants were collected using a pretested structured questionnaire. Clinical characteristics of the patient were collected using a data extraction sheet from the patient clinical information registration sheet. The blood sample was collected from the study participants following standard operating procedures (SOPs) by qualified laboratory personnel after informed consent has been obtained. Following an aseptic vein puncture, about 5 ml of fasting blood sample was collected from each study participant by an experienced laboratory technologist with a sterile disposable syringe. About 2ml of blood sample was used for fasting blood glucose determination and 3 ml of whole blood was dispensed into an ethylenediaminetetraacetic acid (EDTA) test tube for complete blood count analysis. Hematological parameters were determined using DIRUI BF 6500 automated hematology analyzer (DIRUI INDUSTRIAL CO. LTD., P.R., CHINA).

## Data quality management

The blood sample was collected and processed according to the SOPs. Then, blood samples were properly mixed and homogenized by inverting 8–10 times and safety procedures and specimen handling procedures were strictly followed. The performance of the automated hematology analyzer was maintained through the daily running of three-level controls (low level, medium level, and high level) quality control materials. Furthermore, daily background checking was carried out routinely. Automated analyzers and other equipment were cleaned daily before leaving the laboratory as a routine day-to-day activity.

## Data management, statistical analysis, and interpretation

Data were coded, entered, and cleaned using Epi Data 3.1 version and then exported to statistical package for social sciences (SPSS) version 23.0 (IBM Corporation, Armonk, NY, USA). The Kolmogorov-Smirnov and Shapiro Wilk test was conducted to check the distributions of continuous variables. Levene's test was conducted to check the homogeneity of variance. The Independent-T test was used to compare the mean and standard deviation (SD) of hematological parameters between T2DM patients and the control group. Pearson correlation test was used to determine the correlation between fasting blood glucose (FBG) level, body mass index (BMI), systolic blood pressure (SBP), diastolic blood pressure (DBP), and hematological parameters among T2DM patients. Multiple logistic regressions were used to assess the association between independent variables and anemia among T2DM patients. The results were summarized and presented using tables and texts. P-value $< 0.05$ was considered as statistically significant.

## Ethical approval and consent to participate

Ethical approval was obtained from the research and ethics review committee of the College of Medicine and Health Sciences, Wollo University. A support letter was secured from Dessie zonal health office and a permission letter was obtained from the clinical director of Dessie comprehensive specialized hospital. Written informed consent was taken from each study participant before conducting the investigation.

# Results

## Socio-demographic characteristics of study participants

In this study, a total of 240 adult study participants were included. About 120 study participants were included in the T2DM patient group and the remaining 120 study participants were included in the healthy control group. The mean and standard deviation of the age of T2DM patients and the healthy control group were 38.75 (±10.58) and 37.70 (± 9.94) years respectively. Out of the study participants, 64 (53.3%) T2DM patients and 62 (51.7%) of the control group were males. Moreover, about 75 (62.5%) and 98 (81.7%) study participants that were included in T2DM patients and the healthy control group respectively were living in the urban area. Regarding the age classification, 90 (75%) of T2DM patients and 85 (70.8%) of healthy controls were grouped in the age of < 45 years (Table 1).

## Clinical and anthropometric variables among T2DM patients

In this study, a total of 120 T2DM study participants were involved. Out of the diabetic study participants, about 12 (10%) have had visual disturbance, and 15 (12.5%) have had foot ulcers, diabetic ketoacidosis, and neuropathy respectively. About 20 (16.7%) of T2DM patients have had BMI between 25 and 29.9 Kg/m$^2$. Regarding the duration of the disease, about 65 (54.2%) of T2DM patients have had a duration of more than 5 years (Table 2).

**Table 1. Sociodemographic characteristics of study participants at Dessie comprehensive specialized hospital, Northeast Ethiopia from January to June 2021 (n = 240).**

| Variables | Categories | T2DM patients | Healthy controls |
|---|---|---|---|
| | | N (%) | N (%) |
| Age (years) | < 45 | 90 (75.0) | 85 (70.8) |
| | ≥ 45 | 30 (25.0) | 35 (29.2) |
| Gender | Male | 64 (53.3) | 62 (51.7) |
| | Female | 56 (46.7) | 58 (48.3) |
| Residence | Urban | 75 (62.5) | 98 (81.7) |
| | Rural | 45 (37.5) | 22 (18.3) |
| Educational status | Not read and write | 27 (22.5) | 8 (6.7) |
| | Primary school | 36 (30.0) | 8 (6.7) |
| | Secondary school | 27 (22.5) | 20 (16.7) |
| | Diploma and above | 23 (19.2) | 46 (38.3) |
| Occupational status | Student | 20 (16.6) | 14 (11.6) |
| | Non-employed worker | 65 (54.2) | 71 (59.2) |
| | Employed worker | 35 (29.2) | 35 (29.2) |

## Comparison of hematological parameters between T2DM patients and healthy controls

In the current study, the independent T-test was used to compare the mean and the SD of hematological parameters between T2DM patients and the healthy control group. The findings showed that the mean and SD of monocyte count, basophil count, monocyte %, basophil %, RBC count, Hct, MCV, MCH, RDW-SD, MPV, PDW, PLC-R, and plateletcrit had statistically significant difference between T2DM and healthy control group (Table 3).

## Correlation of hematological parameters with anthropometric variables (BMI, DBP, and SBP), and FBG among T2DM patients

In this study, the Pearson correlation test was used to determine the correlation between BMI, DBP, SBP, FBG, and hematological parameters among T2DM patients. Pearson correlation coefficient showed that the total WBC count, neutrophil count, monocyte count, basophil count, RDW-CV, PDW, MPV, PLC-R, and plateletcrit were statistically positively correlated with FBG whereas RBC count, Hgb level, Hct, MCV, MCH, and RDW-SD were statistically negatively correlated with FBG in T2DM patients. Moreover, total WBC count, neutrophil count, monocyte count, basophil count, Hgb, and plateletcrit were statistically positively correlated with BMI while RBC count, Hgb, Hct, MCV, MCH, and RDW-SD were statistically negatively correlated with BMI among T2DM patients (P<0.05). On the other hand, DBP was significantly positively correlated with platelet count and RDW-CV whereas SBP also significantly positively correlated with total WBC count, neutrophil count, basophil count, and PDW. Besides, DBP and SBP showed statistically significant negative correlations with RBC count, Hgb level, and Hct among T2DM patients (Table 4).

## Association of socio-demographic and clinical variables with anemia among T2DM patients

In this study, multivariate logistic regression was used to assess the association of socio-demographic and clinical variables with anemia among T2DM patients. Multivariate logistic regression showed that being non-employed worker (AOR: 3.6, 95% CI, 1.4–46.0, P = 0.002),

**Table 2. Clinical and anthropometric variables of T2DM patients at Dessie comprehensive specialized hospital, Northeast Ethiopia from January to June 2021 (n = 120).**

| Variables | Categories | N (%) |
|---|---|---|
| Diabetic ketoacidosis | Yes | 15 (12.5) |
| | No | 105 (87.5) |
| Foot ulcer | Yes | 15 (12.5) |
| | No | 105 (87.5) |
| HHS | Yes | 13 (10.8) |
| | No | 107 (89.2) |
| Neuropathy | Yes | 15 (12.5) |
| | No | 105 (87.5) |
| Visual disturbance | Yes | 12 (10.0) |
| | No | 108 (90.0) |
| SBP (mmHg) | <120 | 59 (49.2) |
| | 120–139 | 56 (46.7) |
| | ≥140 | 5 (4.2) |
| DBP (mmHg) | <80 | 64 (53.3 |
| | 80–89 | 51 (42.5) |
| | ≥90 | 5 (4.2) |
| BMI (Kg/m$^2$) | <18.5 | 1 (0.8) |
| | 18.5–24.9 | 99 (82.5) |
| | 25–29.9 | 20 (16.7) |
| Treatment regimen | Insulin | 60 (50.0) |
| | Oral hypoglycemic agents | 56 (46.7) |
| | Mixed regimen (insulin and oral agents) | 4 (3.3) |
| Duration of the diseases (years) | < 5 | 55 (45.8) |
| | ≥ 5 | 65 (54.2) |

**Abbreviations:** T2DM; type 2 diabetes mellitus, HHS; hyperosmolar hyperglycemic state, SBP; systolic blood pressure, DBP; diastolic blood pressure, BMI; body mass index

presence of neuropathy (AOR: 13.40, 95%CI, 6.83–26.28, P = 0.00), and duration of the diseases ≥ five years (AOR = 3.2, 95% CI, 1.2–15.3, P = 0.03) were significantly associated and independent predictors for anemia among T2DM patients (Table 5).

## Discussion

Patients with diabetes mellitus show a significant derangement in various hematological parameters [17]. Poorly controlled diabetes is associated with multiple disorders including metabolic, cellular, and blood disturbances leading to vascular complications such as nephropathy, retinopathy, and neuropathy as well as the development of various abnormal metabolic processes that may generate oxidative stress. The released oxidative stress due to elevated blood glucose levels resulted in damaging the various organs, vascular endothelium, and hematological and immunological parameters [5].

In our study, independent T-test showed that the mean and standard deviation of monocyte count, basophil count, monocyte %, basophil %, RBC count, Hct, MCV, MCH, RDW-SD, MPV, PDW, PLC-R, and plateletcrit were statistically significantly different between T2DM patients and healthy control group (P <0.05). This was in concurrent findings reported in Nigeria, India, and Bosnia and Herzegovina [11, 18, 19]. In this study, the mean monocyte count, basophil count, monocyte percent, and basophil percent, MPV, PDW, PLC R, and

**Table 3. Comparison of hematological parameters betweenT2DM patients and healthy controls at Dessie comprehensive specialized hospital, Northeast Ethiopia from January to June 2021 (n = 240).**

| Variables | T2DM patients | Healthy controls | P-value |
|---|---|---|---|
| | Mean ± (SD) | Mean ± (SD) | |
| Total WBC count ($10^9$/L) | 6.84 (2.5) | 6.46 (1.60) | 0.165 |
| Neutrophil count ($10^9$/L) | 3.96 (2.27) | 3.67 (1.34) | 0.226 |
| Lymphocyte count ($10^9$/L) | 2.14 (0.67) | 2.22 (0.51) | 0.282 |
| Monocyte count ($10^9$/L) | 0.51 (0.18) | 0.40 (0.20) | 0.000* |
| Eosinophil count ($10^9$/L) | 0.19 (0.24) | 0.15 (0.10) | 0.130 |
| Basophil count ($10^9$/L) | 0.04 (0.03) | 0.02(0.02) | 0.000* |
| Neutrophil % | 55.07 (13.04) | 55.08 (10.21) | 0.994 |
| Lymphocyte % | 33.62 (11.55) | 35.97 (10.28) | 0.097 |
| Monocyte % | 7.94 (2.73) | 6.14 (2.43) | 0.000* |
| Eosinophil % | 2.74 (2.72) | 2.44 (1.70) | 0.303 |
| Basophil % | 0.55 (0.30) | 0.35 (0.30) | 0.000* |
| RBC count ($10^{12}$/L) | 4.89 (0.90) | 5.31 (0.44) | 0.000* |
| Hgb (g/dL) | 13.65 (2.37) | 15.39 (1.49) | 0.000* |
| Hematocrit (%) | 41.08 (7.36) | 46.31 (4.60) | 0.000* |
| MCV (fL) | 81.20 (4.80) | 86.57 (6.34) | 0.000* |
| MCH(Pg) | 27.31 (2.12) | 28.82 (1.95) | 0.000* |
| MCHC (g/dL) | 335.45 (15.62) | 330.20 (30.38) | 0.093 |
| Platelet count ($10^9$/L) | 260.30 (103.66) | 257.15 (61.82) | 0.776 |
| RDW-SD (fL) | 40.72 (2.35) | 62.88 (88.78) | 0.007* |
| RDW-CV (%) | 13.40 (1.20) | 13.10 (0.58) | 0.09 |
| PDW (fL) | 17.12 (2.77) | 15.35 (1.75) | 0.000* |
| MPV (fL) | 10.18 (1.31) | 9.27 (0.68) | 0.000* |
| PLC-R (%) | 19.70 (3.94) | 17.81 (2.05) | 0.000* |
| Plateletcrit (%) | 0.28 (0.18) | 0.24 (0.56) | 0.007* |
| FBG (mg/dL) | 264.0 (94.90) | 87.97 (8.45) | 0.000* |

**Abbreviations:** WBC; white blood cells, SD; standard deviation, RBC; red blood cells, Hgb; hemoglobin, Hct; hematocrit, MCV; mean corpuscular volume, MCH; mean corpuscular hemoglobin, MCHC; mean corpuscular hemoglobin concentration, RDW-SD, red cell distribution width standard deviation, RDW-CV; red cell distribution width coefficient of variation, PDW; platelet distribution width, MPV; mean platelet volume, PLC-R; platelet large cell ratio; FBG; fasting blood glucose; NB: * = P-value <0.05, statistically significant

plateletcrit were statistically significantly high in the diabetes group as compared to healthy controls. This result was similar to the findings reported in India, Ethiopia, and Turkey in which monocyte count, MPV, and PDW, and in India, the MPV, PDW, and PLC-R were statistically significantly high in diabetes as compared to healthy controls [18, 20–22]. The possible explanation might be persistent hyperglycemia has a direct relationship with the non-enzymatic glycation of various proteins and the increased expression of pro-inflammatory cytokines such as IL-6, and TNF- α in blood circulation. Thus, the elevation of pro-inflammatory cytokines plays an essential role in the development of insulin resistance as well as the changes in the sensitivity of progenitors to erythropoietin (erythroid growth factor). The above inflammatory process may promote apoptosis of immature erythrocytes causing a decrease in the number of circulating erythrocytes consequently causing a reduction in circulating hemoglobin [23–25].

In contrast, the current finding was in discordance results reported in Nigeria where monocyte %, basophil count, and basophil %, and in India the MPV and PDW were significantly

**Table 4. Correlation of hematological parameters with anthropometric variables (BMI, DBP, and SBP), and FBG among T2DM patients at Dessie comprehensive specialized hospital, Northeast Ethiopia from January to June 2021 (n = 120).**

| Variables | FBG | | BMI | | DBP | | SBP | |
|---|---|---|---|---|---|---|---|---|
| | Correlation coefficient (r) | P-value | Correlation coefficient (r) | P–value | Correlation coefficient (r) | P- value | Correlation coefficient (r) | P- value |
| Total WBC count ($10^9$/L) | 0.20 | 0.002* | 0.309 | 0.000* | -0.014 | 0.826 | 0.285 | 0.000* |
| Neutrophil count ($10^9$/L) | 0.199 | 0.002* | 0.333 | 0.000* | 0.023 | 0.721 | 0.286 | 0.000* |
| Lymphocyte count ($10^9$/L) | -0.042 | 0.515 | -0.052 | 0.426 | -0.111 | 0.087 | 0.041 | 0.53 |
| Monocyte count $10^9$/L | 0.26 | 0.001* | 0.146 | 0.023* | -0.016 | 0.81 | 0.099 | 0.125 |
| Eosinophil count $10^9$/L) | 0.108 | 0.095 | 0.123 | 0.058 | -0.039 | 0.549 | 0.106 | 0.101 |
| Basophil count $10^9$/L) | 0.416 | 0.000* | 0.334 | 0.000* | 0.100 | 0.121 | 0.323 | 0.000* |
| RBC count ($10^{12}$/L) | -0.330 | 0.000* | -0.197 | 0.002* | -0.142 | 0.028* | -0.137 | 0.035* |
| Hgb (g/dL) | -0.445 | 0.000* | 0.284 | 0.000* | -0.180 | 0.005* | -0.191 | 0.003* |
| Hct (%) | -0.417 | 0.000* | -0.241 | 0.000* | -0.174 | 0.007* | -0.169 | 0.009* |
| MCV (fL) | -0.417 | 0.000* | -0.226 | 0.000* | 0.107 | 0.098 | -0.51 | 0.431 |
| MCH(Pg) | -0.360 | 0.000* | -0254 | 0.000* | -0.094 | 0.147 | -0.065 | 0.313 |
| MCHC(g/dL) | 0.071 | 0.272 | -0.093 | 0.149 | 0.20 | 0.755 | 0.001 | 0.987 |
| Platelet count ($10^9$/L) | -0.20 | 0.76 | 0.077 | 0.237 | 0.179 | 0.005* | 0.067 | 0.301 |
| RDW-SD (fL) | -0.149 | 0.021* | -0.159 | 0.014* | -0.008 | 0.905 | -0.023 | 0.719 |
| RDW-CV (%) | 0.172 | 0.008* | 0.112 | 0.08 | 0.292 | 0.000* | 0.078 | 0.226 |
| PDW (fL) | 0.276 | 0.000* | 0.098 | 0.130 | 0.084 | 0.192 | 0.144 | 0.026* |
| MPV(fL) | 0.306 | 0.000* | 0.037 | 0.568 | 0.003 | 0.958 | 0.107 | 0.098 |
| PLC-R (%) | 0.187 | 0.004* | -0.044 | 0.50* | -0.035 | 0.591 | 0.090 | 0.166 |
| Plateletcrit (%) | 0.158 | 0.014* | 0.127 | 0.04* | 0.108 | 0.095 | 0.107 | 0.098 |

reduced in T2DM patients as compared to healthy controls [11, 26]. On the other hand, the mean of Hct, MCV, MCH, and RDW-SD was statistically significantly low in T2DM patients as compared to the healthy control group. This result was similar to the findings reported in India, Brazil, Saudi Arabia, Sudan, and Ethiopia in which MCH, MCHC, hematocrit, and RDW-SD were significantly reduced in diabetes compared to healthy controls [9, 27–30]. In diabetic patients, multiple risk factors such as hyperglycemia, hyperosmolarity, oxidative stress, inflammation, and lipid metabolic disorder may affect RBCs metabolism as they may increase aggregation, reduce cell deformability, and reduce membrane fluidity. Consequently, the overall alteration reduces the survival rate, morphology, size, and physiological functions of erythrocytes. Eventually, the overall process affects the physiological functions of RBCs which may in turn aggravates diabetic complications [31]. In contrast, a study conducted in Saudi Arabia showed that MCV, MCH, and MCHC were high in diabetes compared to the controls [28]. The reason for this variation might be due to variation in sample size, socio-economic status, geographical location, laboratory diagnostic method used, and variation in the study populations.

The present finding showed that the total WBC count, neutrophil count, monocyte count, basophil count, RDW-CV, PDW, MPV, PLC-R, and plateletcrit were statistically positively correlated with FBG in T2DM patients. This finding was consistent in studies done in Israel and Poland where FBG levels proportionally elevated for every increment of WBC count [32, 33]. Similarly, other concordant findings were reported in South Korea, India, and Turkey where the MPV, PDW, and P-LC-R showed a significant increase in poor glycemic control

**Table 5. Association of socio-demographic and clinical variables with anemia among T2DM patients at Dessie comprehensive specialized hospital, Northeast Ethiopia from January to June 2021 (n = 120).**

| Variables | Categories | Anemia | | AOR | 95% CI | P-value |
|---|---|---|---|---|---|---|
| | | Yes | No | | | |
| | | n (%) | n (%) | | | |
| Age (years) | ≥ 45 | 13 (10.8) | 17 (14.2) | 1 | | |
| | < 45 | 18 (15.0) | 72 (60.0) | 2.7 | 0.2–37.5 | 0.46 |
| Gender | Male | 11 (9.2) | 53 (44.2) | 0.35 | 0.10–2.14 | 0.25 |
| | Female | 20 (16.7) | 36 (30.0) | 1 | | |
| Residence | Urban | 19 (15.8) | 56 (46.7) | 1.25 | 0.21–7.34 | 0.80 |
| | Rural | 12 (10.0) | 33 (27.5) | 1 | | |
| Educational status | Not read and write | 5 (4.2) | 22 (18.3) | 1.1 | 0.4–11.0 | 0.80 |
| | Primary school | 12 (10.0) | 24 (20.0) | 6.2 | 0.7–95.0 | 0.20 |
| | Secondary school | 8 (6.7) | 19 (15.8) | 1.2 | 0.3–9.8 | 0.90 |
| | Diploma and above | 6 (5.0) | 24 (20.0) | 1 | | |
| Occupational status | Student | 6 (5.0) | 14 (11.7) | 0.87 | 0.5–14.2 | 0.92 |
| | Non-employed worker | 21 (17.5) | 44 (36.7) | 3.6 | 1.4–46.0 | 0.002* |
| | Employed worker | 4 (3.3) | 31 (25.8) | 1 | | |
| Diabetic ketoacidosis | Yes | 9 (7.5) | 6 (5.0) | 0.7 | 0.1–4.3 | 0.20 |
| | No | 22 (18.3) | 83(69.2) | 1 | | |
| Foot ulcer | Yes | 8 (6.7) | 7 (5.8) | 4.3 | 0.3–48.0 | 0.23 |
| | No | 23 (19.2) | 82 (68.3) | 1 | | |
| HHS | Yes | 7 (5.8) | 6 (5.0) | 8.0 | 0.5–126.4 | 0.08 |
| | No | 28 (23.4) | 79 (65.8) | 1 | | |
| Neuropathy | Yes | 9 (7.5) | 6 (5.0) | 13.4 | 6.83–26.28 | 0.00* |
| | No | 22 (18.3) | 83 (69.2) | 1 | | |
| Visual disturbance | Yes | 7 (5.8) | 5 (4.2) | 1.8 | 0.3–6.5 | 0.45 |
| | No | 25 (20.8) | 83 (69.2) | 1 | | |
| SBP (mmHg) | ≤120 | 11 (9.2) | 48 (40.0) | 0.33 | 0.01–6.32 | 0.46 |
| | >120 | 20 (16.7) | 41 (34.2) | 1 | | |
| DBP (mmHg) | ≤80 | 12 (10.0) | 52 (43.3) | 1.5 | 0.07–31.8 | 0.78 |
| | >80 | 19 (15.8) | 37 (30.8) | 1 | | |
| BMI (Kg/m$^2$) | <25 | 23 (19.2) | 78 (65.0) | 2.36 | 0.3–17.1 | 0.10 |
| | ≥25 | 8 (6.7) | 11 (9.2) | 1 | | |
| Medication regimen | Insulin | 15 (12.5) | 45 (37.5) | 3.66 | 0–384 | 0.78 |
| | Oral hypoglycemic agents | 15 (12.5) | 41 (34.2) | 6.33 | 0.1–661 | 0.69 |
| | Insulin and oral hypoglycemic agents | 1 (0.8) | 3 (2.5) | 1 | | |
| Duration the of diseases (years) | <5 | 6 (5.0) | 49 (40.8) | 1 | | |
| | ≥5 | 25 (20.8) | 40 (33.3) | 3.2 | 1.2–15.3 | 0.03* |

**Abbreviations:** AOR: Adjusted Odds Ratio; CI: confidence interval

diabetes patients [14, 34, 35]. On the other hand, discordant findings were reported in India where MPV showed a negative correlation with FBG in diabetes [26]. In diabetes, hyperglycemia, dyslipidemia, insulin resistance, and oxidative stress could stimulate the production of pro-inflammatory cytokines, activation of inflammatory signaling pathways, and recruitment of immune cells that can contribute to the elevated level of white blood cells and its sub-population [33]. Moreover, Hyperglycemia, insulin resistance, and insulin deficiency contribute to increased platelet reactivity through direct effects via promoting glycation of platelet proteins leading to morphological and functional alteration of platelet indices [35].

In this study, FBG was statistically negatively correlated with RBC count, Hgb, Hct, MCV, MCH, and RDW-SD among T2DM patients. This finding was similar to studies done in Saudi Arabia and India where the FBG level was negatively correlated with RDW-SD and MCV among T2DM patients [28, 36]. Moreover, this result was in opposite finding reported in Japan where RBC and RDW-SD were positively correlated with FBG in diabetes [37]. In the current study, total WBC count, neutrophil count, monocyte count, basophil count, Hgb, and plateletcrit were statistically positively correlated with BMI. This was similar in finding reported in India where the total WBC count and Hgb level, and in Turkey in which the total WBC count and the neutrophil count were positively correlated with BMI inT2DM patients [21, 36].

In the current study, RBC count, Hgb, Hct, MCV, MCH, and RDW-SD were statistically negatively correlated with BMI. This result was in concurrent findings done in Brazil in which the RBC count and hematocrit were negatively correlated with BMI in T2DM patients [27]. In this study, DBP was significantly positively correlated with platelet count and RDW-CV whereas SBP also significantly positively correlated with total WBC count, neutrophil count, basophil count, and PDW among T2DM patients. On the other hand, DBP and SBP showed significant negative correlations with RBC count, Hgb, and Hct among T2DM. This was in agreement with the findings reported in India and Turkey in which the total WBC count, neutrophil count, basophil count, and PDW were positively correlated with DBP and SBP among T2DM [21, 36].

In the present study, the overall prevalence of anemia was 31 (25.8%) among T2DM patients and a higher prevalence of anemia (16.7%) was found in female patients. This was inline in findings reported in Australia (23%), and Ethiopia (24.8%) among diabetes [38, 39]. On the other hand, the current prevalence was low compared to the findings reported in Brazil (34.2%), Ethiopia (30.2%), and Pakistan (63%) [27, 40, 41]. In contrast, this finding was high compared to the result found in Eastern Ethiopia (20.1%), and Gondar, Ethiopia (8.06%) among T2DM patients [42, 43]. In diabetess, prolonged exposure to hyperglycaemia is directly associated with cardiovascular disease and renal disease. Therefore, several factors which have been implicated in the development of anemia in diabetes include erythropoietin deficiency, iron deficiency, decreased lifespan of red blood cells, chronic blood loss, chronic inflammation, oxidative stress, nutritional deficiency, and chronic suppression of erythropoiesis [44, 45].

Multivariate logistic regression showed that being non-employed worker (AOR: 3.6, 95% CI, 1.4–46.0, P = 0.002), presence of neuropathy (AOR: 13.40, 95%CI, 6.83–26.28, P = 0.00), and duration of the diseases ≥ 5years (AOR = 3.2, 95% CI, 1.2–15.3, P = 0.03) were significantly associated and independent predictors of anemia in T2DM patients. This finding was in line with the findings reported in Ethiopia in which the duration of diabetes ≥ 5 years (AOR = 1.9, 95% CI: 1.0, 3.7) and having diabetic complications were significantly associated with anemia [40, 42]. In Malaysia and Nigeria, diabetes patients who had diabetic complications (AOR = 3.12, 95% CI: 1.51 to 6.46, P = 0.002) and longer duration of diabetes (AOR = 2.1, 95% CI: 1.04–4.25, P = 0.03) were significantly associated with anemia among T2DM patients [46, 47]. The reason might be that longer exposure to hyperglycemia is significantly associated with the generation of reactive oxygen species that may be related to multiple organ damage possibly making the diabetic patients at higher risk for developing anemia [48, 49]. Moreover, hyperglycemia, hyperinsulinemia, dyslipidemia, insulin resistance syndrome, inflammation, dysglycemia, oxidative stress, inflammation, and endothelial dysfunction are significantly associated with the development of both micro vascular and macro vascular complications including neuropathy [50].

Diabetic neuropathy is a life-threatening complication that involves both peripheral and autonomic nerves. Similarly, like other micro vascular complications, the risk of developing diabetic neuropathy is proportional to both the magnitude and duration of hyperglycemia, and some individuals may possess genetic attributes that affect their predisposition which may increase its complications. The precise cause of injury to the peripheral nerves from hyperglycemia is not known but is likely related to mechanisms such as polyol accumulation, injury from the advanced glycation end product, and oxidative stress that leads to peripheral neuropathy including sensory, focal, and autonomic neuropathies [51–53]. The study has faced some limitations such as being cross-sectional nature of the study did not show the cause-effect relationship between diabetic complications and hematological alteration in the T2DM patients. Hemoglobin A1C or glycated hemoglobin was not determined which is the best diagnostic marker to assess the development of micro and macro vascular complications due to accessibility and availability constraints. Patients with nephropathic complications did not include in the study.

## Conclusion

In this study, the mean and standard deviation of the main hematological parameters showed a statistically significant difference between T2DM patients and healthy controls. Furthermore, FBG was statistically negatively correlated with some hematological parameters and statistically positively correlated with other hematological parameters. Anthropometric variables such as BMI, SBP, and DBP were statistically positively correlated with some hematological parameters and statistically negatively correlated with other hematological parameters among T2DM patients. The overall prevalence of anemia was 25.8% in T2DM patients and a higher prevalence of anemia (16.7%) was found among female patients. Multivariate logistic regression showed that being a non-employed worker, presence of neuropathy, and the duration of the disease ≥ 5 years had a significant association with anemia in T2DM patients. Patients with T2DM are associated with significant alterations in various hematological parameters. Therefore, hematological parameters should be routinely determined among T2DM patients for proper management and reduction of further complications.

## Supporting information

**S1 File.**
(DOCX)

**S1 Data.**
(XLSX)

**S2 Data.**
(XLSX)

## Acknowledgments

We would like to thank the study participants and data collectors who participated in this study. Furthermore, we would also give our heartfelt thanks to all who support and stood by our side during the study.

## Author Contributions

**Conceptualization:** Hussen Ebrahim, Temesgen Fiseha, Yesuf Ebrahim, Habtye Bisetegn.

**Data curation:** Hussen Ebrahim, Yesuf Ebrahim, Habtye Bisetegn.

**Formal analysis:** Hussen Ebrahim, Temesgen Fiseha, Habtye Bisetegn.

**Investigation:** Hussen Ebrahim, Yesuf Ebrahim.

**Methodology:** Hussen Ebrahim, Temesgen Fiseha, Yesuf Ebrahim, Habtye Bisetegn.

**Project administration:** Hussen Ebrahim.

**Resources:** Hussen Ebrahim.

**Software:** Hussen Ebrahim, Habtye Bisetegn.

**Supervision:** Hussen Ebrahim, Yesuf Ebrahim.

**Validation:** Hussen Ebrahim, Temesgen Fiseha, Yesuf Ebrahim, Habtye Bisetegn.

**Visualization:** Hussen Ebrahim, Temesgen Fiseha, Yesuf Ebrahim.

**Writing – original draft:** Hussen Ebrahim, Temesgen Fiseha, Yesuf Ebrahim, Habtye Bisetegn.

**Writing – review & editing:** Hussen Ebrahim.

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
