## [Decision Letter · Decision Letter 0]

5 Apr 2022

PONE-D-21-30534Comparison of Hematological parameters between Type 2 Diabetes Mellitus and Healthy Controls at Dessie Comprehensive Specialized Hospital, Northeast Ethiopia: Comparative Cross-sectional studyPLOS ONE

Dear Dr. Hussen Ebrahim Adem,

Thank you for submitting your manuscript to PLOS ONE. After careful consideration, we feel that it has merit but does not fully meet PLOS ONE’s publication criteria as it currently stands. Therefore, we invite you to submit a revised version of the manuscript that addresses the points raised during the review process.

Please address all points raised by reviewers.

We look forward to receiving your revised manuscript.

Kind regards,

Paolo Magni

Academic Editor

PLOS ONE

Journal Requirements:

Additional Editor Comments (if provided):

Please reply to all reviewer's comments.

Reviewers' comments:

Reviewer's Responses to Questions

**Comments to the Author**

1. Is the manuscript technically sound, and do the data support the conclusions?

Reviewer #1: No

2. Has the statistical analysis been performed appropriately and rigorously? 

Reviewer #1: I Don't Know

3. Have the authors made all data underlying the findings in their manuscript fully available?

Reviewer #1: Yes

4. Is the manuscript presented in an intelligible fashion and written in standard English?

Reviewer #1: No

5. Review Comments to the Author

Reviewer #1: I consider that the work done by Hussen and coworkers is important because they are trying to exploit data acquired from the hematological test that are a useful available test, but the document needs to be improved. It seems that the main objective of the study was set after the authors concluded the statistical analysis, this comment is based on the lack of sufficient information regarding anemia in the population studied. In the introduction, the authors also indicate that there are commonly two types of diabetes, according to ADA there are 4 general categories, see American Diabetes Association Professional Practice Committee; 2. Classification and Diagnosis of Diabetes: Standards of Medical Care in Diabetes—2022. Diabetes Care 1 January 2022; 45 (Supplement_1): S17–S38. https://doi.org/10.2337/dc22-S002. The authors also said that T1DM affects commonly children and adolescents and T2DM affects young adults. This statement is no longer accurate, as both diseases occur in both age groups (see the reference above). I consider that reference 4 does not support the information about the diabetes classification. There is also a 2021 version of the IDF atlas, I think it is a good idea to actualize the reference information.

Study participants: The authors did not make an appropriate description of the clinical characteristics of the population studied. They should include the inclusion criteria. Moreover, they need to include the values of FBG, HbA1c and explain how they evaluated the existence of diabetic complications it is hard to believe that the patients presented neuropathy, hyperosmolar hyperglycemia, foot ulcers, etc. and they do not present a certain degree of renal dysfunction, it is well known that renal dysfunction is a cause of anemia. Few patients reported those kinds of complications, if the authors exclude those patients, will they get the same results?.

Sample size: The authors should include the statistical formulae used to estimate the sample size and which variables they took into account to perform the estimation of the sample size and support the information with reference values. Concerning the statistical analysis, I do not have the expertise to consider them.

Figures and tables: The authors need to detail the format presentation of the variables, Table 1 is not well introduced in the text; furthermore, line one of this table lacks a variable title, what does < 45 mean? I suggest including the age range.

Authors should uniform the title of the variables in the tables, sometimes they write with uppercase and in other places of the document with lowercase. There is a format error in table 2.

In the discussion section, they did not argue about the limitations of the study nor the strengths, and in the conclusion section, the authors seem to summarize the results.

References should be written according to the requirements of the journal.

6. PLOS authors have the option to publish the peer review history of their article (what does this mean?). If published, this will include your full peer review and any attached files.

Reviewer #1: No

---

## [Author Response · Author response to Decision Letter 0]

3 May 2022

Response is submitted as response to editor and response to reviewer files

---

## [Decision Letter · Decision Letter 1]

30 May 2022

PONE-D-21-30534R1Comparison of Hematological parameters between Type 2 Diabetes Mellitus and Healthy Controls at Dessie Comprehensive Specialized Hospital, Northeast Ethiopia: Comparative Cross-sectional studyPLOS ONE

Dear Dr. Hussen Ebrahim Adem,

Thank you for submitting your manuscript to PLOS ONE. After careful consideration, we feel that it has merit but does not fully meet PLOS ONE’s publication criteria as it currently stands. Therefore, we invite you to submit a revised version of the manuscript that addresses the points raised during the review process.

We look forward to receiving your revised manuscript.

Kind regards,

Paolo Magni

Academic Editor

PLOS ONE

Journal Requirements:

Additional Editor Comments:

Some comments still need to be addressed properly.

Reviewers' comments:

Reviewer's Responses to Questions

**Comments to the Author**

1. If the authors have adequately addressed your comments raised in a previous round of review and you feel that this manuscript is now acceptable for publication, you may indicate that here to bypass the “Comments to the Author” section, enter your conflict of interest statement in the “Confidential to Editor” section, and submit your "Accept" recommendation.

Reviewer #1: (No Response)

2. Is the manuscript technically sound, and do the data support the conclusions?

Reviewer #1: Partly

3. Has the statistical analysis been performed appropriately and rigorously? 

Reviewer #1: I Don't Know

4. Have the authors made all data underlying the findings in their manuscript fully available?

Reviewer #1: Yes

5. Is the manuscript presented in an intelligible fashion and written in standard English?

Reviewer #1: No

6. Review Comments to the Author

Reviewer #1: The authors improve the manuscript significantly, but there are some things to attend

Introduction.- Even if the authors did not reply to my questions they took into account my suggestions actualizing the references about the classification of diabetes and the reference.

Study population.- The population description was improved but there is a lack of some important conditions (my humble point of view), such as glycated hemoglobin to state the glycemic control of the population. This is important because of the macro and microvascular complications

Sample size.- The authors did not justify your strategy of “rule of the thumb”

Concerning the statistical analysis, I do not have the expertise to consider them.

Figures and tables.- In table 1, the authors changed the number of subjects that were under 45 years old. In the first version, the number was inverted. How do they explain this?

DISCUSSION.- The authors stated that poorly controlled patients are at higher risk of complications such as endothelial dysfunction and an increase in oxidative stress, but they omitted to describe the control degree of the population included in this study. In table 4 the Hgb did not correlates negatively with BMI the correlation was positive (r= 0.284) see table 4. The study of Adane T, Getawa S. Anaemia, and its associated factors among diabetes mellitus patients in Ethiopia: A systematic review and meta-analysis. Endocrinol Diabetes Metab. 2021 May 14;4(3):e00260. doi: 10.1002/edm2.260. found an odds for anemia of 2.98 and a major risk in males >60 years old and who had an illness duration of more than five years. This study found that anemia was more frequent in females than in males and at a younger age. Might the authors explain why? I think the contrasting results are important. In this sense, the conclusion would emphasize the importance of the diabetes follow-up in the early stages of the illness and why this study is relevant, I feel this is still the same as the past version but without numbers.

There are some errors in the references please check the style and numeration.

7. PLOS authors have the option to publish the peer review history of their article (what does this mean?). If published, this will include your full peer review and any attached files.

Reviewer #1: No

---

## [Author Response · Author response to Decision Letter 1]

24 Jun 2022

Available in the attached document

---

## [Decision Letter · Decision Letter 2]

14 Jul 2022

Comparison of Hematological parameters between Type 2 Diabetes Mellitus and Healthy Controls at Dessie Comprehensive Specialized Hospital, Northeast Ethiopia: Comparative Cross-sectional study

PONE-D-21-30534R2

Dear Dr. Ebrahim,

We’re pleased to inform you that your manuscript has been judged scientifically suitable for publication and will be formally accepted for publication once it meets all outstanding technical requirements.

Kind regards,

Muhammad Sajid Hamid Akash

Academic Editor

PLOS ONE

Additional Editor Comments (optional):

Reviewers' comments:

Reviewer's Responses to Questions

**Comments to the Author**

1. If the authors have adequately addressed your comments raised in a previous round of review and you feel that this manuscript is now acceptable for publication, you may indicate that here to bypass the “Comments to the Author” section, enter your conflict of interest statement in the “Confidential to Editor” section, and submit your "Accept" recommendation.

Reviewer #1: All comments have been addressed

2. Is the manuscript technically sound, and do the data support the conclusions?

Reviewer #1: Yes

3. Has the statistical analysis been performed appropriately and rigorously? 

Reviewer #1: I Don't Know

4. Have the authors made all data underlying the findings in their manuscript fully available?

Reviewer #1: Yes

5. Is the manuscript presented in an intelligible fashion and written in standard English?

Reviewer #1: (No Response)

6. Review Comments to the Author

Reviewer #1: The authors have addressed my comments. I think there are still some typographical errors. In some parts of the text, they wrote: "in line" and others "inline". The size of the letter is not uniform. I think the article can be published.

7. PLOS authors have the option to publish the peer review history of their article (what does this mean?). If published, this will include your full peer review and any attached files.

Reviewer #1: No

---

## [Editor Report · Acceptance letter]

18 Jul 2022

PONE-D-21-30534R2 

Comparison of hematological parameters between Type 2 diabetes mellitus patients and healthy controls at Dessie comprehensive specialized hospital, Northeast Ethiopia: Comparative cross-sectional study 

Dear Dr. Ebrahim:

I'm pleased to inform you that your manuscript has been deemed suitable for publication in PLOS ONE. Congratulations! Your manuscript is now with our production department. 

Kind regards, 

on behalf of

Dr. Muhammad Sajid Hamid Akash 

Academic Editor

PLOS ONE